# Distinct Concentration-Dependent Molecular Pathways Regulate Bone Cell Responses to Cobalt and Chromium Exposure from Joint Replacement Prostheses

**DOI:** 10.3390/ijms22105225

**Published:** 2021-05-14

**Authors:** Karan M. Shah, Mark J. Dunning, Alison Gartland, J. Mark Wilkinson

**Affiliations:** 1The Mellanby Centre for Musculoskeletal Research, Department of Oncology and Metabolism, The University of Sheffield, Beech Hill Rd, Sheffield S10 2RX, UK; k.shah@sheffield.ac.uk; 2Sheffield Bioinformatics Core, The University of Sheffield, 385a Glossop Rd, Sheffield S10 2HQ, UK; m.j.dunning@sheffield.ac.uk

**Keywords:** hip replacement, osteoblasts, osteoclasts, cobalt, chromium, gene expression, microarray, prosthesis surface

## Abstract

Systemic cobalt (Co) and chromium (Cr) concentrations may be elevated in patients with metal joint replacement prostheses. Several studies have highlighted the detrimental effects of this exposure on bone cells in vitro, but the underlying mechanisms remain unclear. In this study, we use whole-genome microarrays to comprehensively assess gene expression in primary human osteoblasts, osteoclast precursors and mature resorbing osteoclasts following exposure to clinically relevant circulating versus local periprosthetic tissue concentrations of Co^2+^ and Cr^3+^ ions and CoCr nanoparticles. We also describe the gene expression response in osteoblasts on routinely used prosthesis surfaces in the presence of metal exposure. Our results suggest that systemic levels of metal exposure have no effect on osteoblasts, and primarily inhibit osteoclast differentiation and function via altering the focal adhesion and extracellular matrix interaction pathways. In contrast, periprosthetic levels of metal exposure inhibit both osteoblast and osteoclast activity by altering HIF-1α signaling and endocytic/cytoskeletal genes respectively, as well as increasing inflammatory signaling with mechanistic implications for adverse reactions to metal debris. Furthermore, we identify gene clusters and KEGG pathways for which the expression correlates with increasing Co^2+^:Cr^3+^ concentrations, and has the potential to serve as early markers of metal toxicity. Finally, our study provides a molecular basis for the improved clinical outcomes for hydroxyapatite-coated prostheses that elicit a pro-survival osteogenic gene signature compared to grit-blasted and plasma-sprayed titanium-coated surfaces in the presence of metal exposure.

## 1. Introduction

The observed early failure rate for joint replacement prostheses that use a metal-on-metal (MoM) bearing have highlighted the adverse effects of cobalt (Co) and chromium (Cr) on tissue in the periprosthetic environment [1,2]. Tribocorrosion at the bearing surfaces of MoM prostheses, and at modular junctions of all hip prostheses, causes persistent elevation of Co and Cr levels both locally and systemically.

Clinically, we have reported that patients with MoM hip resurfacings and relatively higher circulating concentrations of Co and Cr (in the interquartile range 1 to 7 µg/L) have reduced bone turnover systemically and increased total bone mineral density compared to individually matched patients with a conventional metal-on-polyethylene (MoP) bearing hip prostheses [3]. There is also extensive evidence that suggests detrimental effects of metal exposure on the survival and function of several cell types including monocytes, macrophages and bone cells in vitro [4,5,6,7,8]. However, the molecular mechanisms for these effects remain unclear.

A recent RNA-seq-based study compared synovial tissues from patients with adverse reaction to metal debris (ARMD) to those from patients with MoP and reported changes in genes associated with redox homeostasis and macrophage activation [9]. Another study that investigated the effects of Co and Cr exposure in osteoblast-like cells using an osteogenic gene-panel, at supra-physiological concentrations (250 µM: ~12,500 µg/L), reported down-regulation of genes involved in TGF-beta signaling and collagen production [10]. While these studies increase our understanding of the effects metal exposure may have in the periprosthetic environment, they fail to provide a comprehensive transcriptome-wide insight into the response of primary human osteoblasts and osteoclasts exposed to clinically equivalent systemic concentrations of metal exposure.

We have previously shown that routinely used topographical and chemical alteration of prosthetic surfaces alter the physiological response of osteoblast-like cells to clinically relevant levels of metal exposure [7]. We observed a smaller reduction in osteogenic activity of SaOS-2 cells following Co and Cr exposure when grown on hydroxyapatite-coated surfaces (HA) compared to those grown on grit-blasted (GB) or plasma-sprayed titanium-coated (Ti) surfaces, with implications for osseointegration of the prosthesis. The molecular mechanisms that mediate this relative protective effect also remains unknown.

In this study, we use whole-genome microarrays to assess differential gene expression in primary human osteoblasts and osteoclasts grown in vitro, and in osteoblasts grown on different prosthesis surfaces following exposure to clinically relevant concentrations of Co and Cr.

## 2. Results

### 2.1. Effect of Cobalt and Chromium Exposure on Primary Human Osteoblasts

For osteoblasts grown in vitro on tissue culture plates and exposed to systemic concentrations of metal ions observed for well-functioning MOM prosthesis, 5 µg/L Co^2+^:Cr^3+^ did not significantly alter gene expression compared to the untreated cells. In contrast, 500 µg/L Co^2+^:Cr^3+^ (equivalent to the concentration of metal ions found periprosthetically for a well-functioning MOM prosthesis) up-regulated 25 genes (Table 1). KEGG pathway analyses identified overrepresentation of genes associated with the HIF-1 signaling pathway (KEGG:04066; *P* = 4.15 × 10^−7^), fructose and mannose metabolism (KEGG:00051; *P* = 8.90 × 10^−4^) and renal cell carcinoma (KEGG:05211; *P* = 3.345 × 10^−2^) in the up-regulated genes (Table 2). Specifically, in the HIF-1 signaling pathway, genes associated with anaerobic metabolism and reduction in oxygen consumption (*PDK-1* and *PFKFB3*; Log_2_FC of 2.04 and 1.59 and *P* values of 1.1 × 10^−7^ and 6.6 × 10^−4^ respectively), and increase in oxygen delivery (*VEGFA*, Log_2_FC = 1.6 and *P* = 3.8 × 10^−4^) were up-regulated. Other known HIF-1 targets such as *ANKRD37* (Log_2_FC = 1.89 and *P* = 3.6 × 10^−4^) and *BNIP3* (Log_2_FC = 1.83 and *P* = 1.4 × 10^−6^) were also up-regulated.

Gene probes that showed significant dose-dependent effects with increasing concentrations of Co^2+^:Cr^3+^ formed nine individual clusters (Figure 1, Appendix A). Functional classification of these gene clusters identified HIF-1 signalling (*P* = 5.2 × 10^−7^; Cluster 1) and Glycolysis/Gluconeogenesis (*P* = 1.02 × 10^−6^; Cluster 2) pathways that associated positively with increasing concentrations. Exposure to CoCr nanoparticles (100 nanoparticles of Co and Cr_2_O_3_ each per cell) did not alter the gene expression significantly.

### 2.2. Effect of Cobalt and Chromium Exposure on Primary Human Osteoclasts

For osteoclast precursors exposed to systemic Co^2+^:Cr^3+^ concentration of 5 µg/L, 313 genes were down-regulated (Appendix A). KEGG pathway analyses identified overrepresentation of genes associated with the focal adhesion pathway (KEGG:04510; *P* = 1.8 × 10^−3^) and ECM-receptor interaction (KEGG:04512; *P* = 6.3 × 10^−3^) among others (Figure 2). We also observed down-regulation of genes associated with osteoclast differentiation (*DCSTAMP* and *GPER1*; Log_2_FC = −1.87 and −1.64, *P* = 2.4 × 10^−3^ and 2.9 × 10^−3^ respectively) and apoptosis (*HRK* and FAS; Log_2_FC =−1.65 and −2.3, *P* = 3.6 × 10^−3^ and 7.3 × 10^−3^). This was accompanied by an increase in genes associated with epigenetic modifications (*KMT5C*, Log_2_FC = −1.53, *P* = 4.5 × 10^−3^) and Ca^2+^ signaling pathways (*ANO3* and *SELENOP*; Log_2_FC = −1.56 and 1.87.3, *P* = 6.4 × 10^−3^ and 9.3 × 10^−3^; Appendix A). 

Exposure to higher concentration of 500 µg/L Co^2+^:Cr^3+^ caused a down-regulation in 467 genes, and an up-regulation in 125 genes (Appendix A). Functional classification identified the down-regulated genes to be overrepresented in the endocytosis pathway (KEGG:04144; *P* = 9.3 × 10^−3^). These include genes involved in both the formation of endosomes (*PIP5K*, Log_2_FC = −1.63, *P* = 2.1 × 10^−3^; and *PLD*, Log_2_FC = −1.59, *P* = 1.75 × 10^−4^) and its trafficking (*EHD2*, Log_2_FC = −1.60, *P* = 1.9 × 10^−3^; and *EHD3*, Log_2_FC = −1.73, *P* = 8.82 × 10^−4^). Up-regulated genes were associated with the cytokine-cytokine receptor interaction pathway (KEGG:04060; *P* = 1.1 × 10^−13^) and TNF signaling pathway (KEGG:04668; *P* = 3.5 × 10^−4^).

Gene probes that showed significant dose-dependent effects with increasing concentrations of Co^2+^:Cr^3+^ formed eight individual clusters (Appendix A). Functional classification of these gene clusters identified calcium signalling (KEGG:04020; *P* = 1.3 × 10^−3^) and Ras signalling pathway (KEGG:04014; *P* = 3.6 × 10^−3^) to be enriched in Cluster 1, which associated negatively with increasing metal concentrations (Figure 3A). Cluster 4 which positively correlates with increasing metal concentrations (Figure 3B) had the cytokine-cytokine receptor interaction pathway (KEGG:04060; *P* = 4.7 × 10^−2^) enriched which included genes such as *IL13RA1* and *IFNGR1* that are known to reduce osteoclast activity [11,12].

For osteoclast precursors exposed to CoCr nanoparticles, we observed down-regulation of 525 genes, including genes that associate with the endocytosis pathway (KEGG:04144; *P* = 1.1 × 10^−3^), and those that regulate actin cytoskeleton (KEGG:04810; *P* = 8.0 × 10^−3^; Appendix A). There were 121 genes up-regulated that were overrepresented in the transcriptional misregulation in cancer pathway (KEGG:05202; *P* = 5.9 × 10^−4^), including chromatin regulator *HMGA2* (Log_2_FC = 1.14; *P* = 2.9 × 10^−3^) and transcription factor *HOXA10* (Log_2_FC = 1.03; *P* = 2.98 × 10^−5^).

For mature osteoclasts exposed to systemic Co^2+^:Cr^3+^ concentration of 5 µg/L, 185 genes were down-regulated (Appendix A). Functional classification identified their association with focal adhesion (KEGG: 04510; *P* = 3.4 × 10^−3^), ECM interaction (KEGG:04512; *P* = 9.5 × 10^−3^) and PI3K-Akt signaling pathways (KEGG:04151; *P* = 4.6 × 10^−2^). We also observed down-regulation of similar genes to those observed in osteoclast precursors, and those that osteoclast differentiation (*DCSTAMP* and *GPER1*; Log_2_FC = −1.74 and −1.66, *P* = 6.0 × 10^−3^ and 4.2 × 10^−3^ respectively).

At higher concentrations of 500 µg/L Co^2+^:Cr^3+^, as with the lower concentration of 5 µg/L, genes from the focal adhesion (KEGG: 04510; *P* = 3.0 × 10^−3^) and PI3K-Akt signaling pathways (KEGG: 04151; *P* = 2.1 × 10^−2^) were down-regulated. Other genes that were associated with osteoclast survival and function that were down-regulated included *CSF1* (Log_2_FC = −2.38; *P* = 3 × 10^−4^) and *ITGBL1* (Log_2_FC = −1.62; *P* = 9 × 10^−3^; Appendix A). Genes known to play a role in osteoblast differentiation such as *WNT5A* (Log_2_FC = 1.42; P = 2.03 × 10^−6^) and *ATF5* (Log_2_FC = 1.97; *P* = 3.1 × 10^−3^) were up-regulated at this concentration (Appendix A).

Gene probes that showed significant dose-dependent effects with increasing concentrations of Co^2+^:Cr^3+^ formed nine individual clusters (Appendix A). Genes from the focal adhesion pathway (KEGG: 04510; P = 5.0 × 10^−2^, Cluster 6) were found to be enriched from a cluster that negatively associated with increasing metal concentrations (Table 3).

Focal adhesion (KEGG: 04510; *P* = 5.4 × 10^−4^) and endocytosis (KEGG:04144; *P* = 2.5 × 10^−3^) were the primary pathways down-regulated in mature osteoclasts exposed to CoCr nanoparticles. Specifically, integrins *ITGA3* (Log_2_FC = −1.03; *P* = 4.9 × 10^−3^) and *ITGB3* (Log_2_FC = −1.89; *P* = 8.9 × 10^−4^), and cytoskeletal genes *PXN* (Log_2_FC = −1.36; *P* = 1.27 × 10^−4^) and *FLNB* (Log_2_FC = −1.41; P = 1.17 × 10^−4^) were down-regulated in the focal adhesion pathway. Endocytosis associated genes included *PIP5K* (Log2FC = −2.1; *P* = 6.3 × 10^−4^), *EHD2* (Log_2_FC = −2.92; *P* = 5.9 × 10^−4^), *EHD3* (Log_2_FC = −1.94; *P* = 5.8 × 10^−4^), and *RAB11FIP1* (Log_2_FC = −1.52; *P* = 4.8 × 10^−4^). *IL6* (Log_2_FC = 1.51; *P* = 7.06 × 10^−4^) and *SELENOP* (Log_2_FC = 2.08; *P* = 2 × 10^−3^) both of which are implicated in osteoclast activity and Ca^2+^ homeostasis were up-regulated (Appendix A).

### 2.3. Effect of Cobalt and Chromium Exposure on Primary Human Osteoblasts on Prosthesis Surfaces

For osteoblasts grown on prosthesis surfaces in the absence of metal exposure, we did not observe significant alteration in gene expression between cells on GB and Ti surfaces. For cells grown on HA surfaces, we observed an up-regulation in 40 genes compared to cells grown on GB surfaces. While no KEGG pathway was identified for these genes, we observed increased expression in *BMP2* (Log_2_FC = 2.6; *P* = 4.71 × 10^−7^) and *SPP1* (Log_2_FC = 2.78; *P* = 3.9 × 10^−4^) which are associated with increased osteoblast differentiation (Appendix A). When comparing cells on HA surfaces to those on Ti surfaces, we observed 69 genes down-regulated on Ti surfaces (Appendix A), of which 12 were common to those expressed at lower levels on GB surfaces including *BMP2* and *SPP1*.

Following metal exposure at periprosthetic concentrations of Co^2+^:Cr^3+^ (1000 µg/L), the expression profile for osteoblasts on GB surfaces was similar to that of cells on Ti surfaces with only three genes differentially regulated. We observed an up-regulation of *TSPAN13* (Log_2_FC = 1.13; *P* = 9.75 × 10^−4^), *GUCY2G* (Log_2_FC = 1.01; *P* = 9.77 × 10^−4^) and *UBE2DNL* (Log_2_FC = 1.02; *P* = 9.79 × 10^−3^) for cells on GB surfaces compared to those on Ti surfaces. In comparison, cells on HA surfaces had 69 probes down-regulated with no specific KEGG pathway enriched for the down-regulated genes (Appendix A). Several genes that were down-regulated on HA surfaces are known to play a vital role in cell cycle progression and mitosis including *CCNB2* (Log_2_FC = −1.3; *P* = 3.5 × 10^−3^), *KIF22* (Log_2_FC = −1.33; *P* = 1.1 × 10^−3^), *CDCA2* (Log_2_FC = 1.10; *P* = 5.2 × 10^−3^) and *CENPN* (Log_2_FC = −1.18; *P* = 5.5 × 10^−3^). Genes that were up-regulated on HA surfaces compared to GB surfaces included those that influence osteoblast differentiation including *BMP2* (Log_2_FC = 2.66; *P* = 2.56 × 10^−6^), *SPP1* (Log_2_FC = 2.61; *P* = 3.09 × 10^−4^) and *OMD* (Log_2_FC = 1.15; *P* = 7.1 × 10^−3^). We also observed genes from the Glycerolipid metabolism pathway (KEGG: 00561; *P* = 7.09 × 10^−3^) to be up-regulated in cells from HA surfaces compared to GB (Appendix A).

In the presence of periprosthetic concentrations of Co^2+^:Cr^3+^ ions (1000 µg/L), osteoblast grown on HA surfaces had higher expression of genes associated with osteoblast differentiation and mineralization compared to cells on Ti surfaces, similar to those observed with GB surfaces. We observed an increase in *BMP2* (Log_2_FC = 2.71; *P* = 2.12 × 10^−6^) and *SPP1* (Log_2_FC = 3.04; *P* = 8.58 × 10^−5^) expression; inorganic pyrophosphate transporter *ANKH* (Log_2_FC = 1.1; *P* = 3.9 × 10^−3^), and extracellular matrix protein *MGP* (Log_2_FC = 2.22; *P* = 7.8 × 10^−3^) which regulate mineralization. *RAB27B*, a gene responsible for vesicular trafficking was also up-regulated on HA surfaces (Log_2_FC = 1.4; *P* = 9.71 × 10^−5^; Appendix A).

Following CoCr nanoparticle exposure, we did not observe large-scale alterations in gene expression between osteoblasts on GB surfaces and Ti surfaces. The only differential expression we observed was a down-regulation in *MX1* (Log_2_FC = −1.39; *P* = 3.7 × 10^−3^) and an up-regulation in *GRIN3A* (Log_2_FC = −1.12; *P* = 1.1 × 10^−3^) expression in cells on GB surfaces compared to Ti surfaces. Genes enriched in the necroptosis pathway (KEGG: 04217; *P* = 1.18*P* × 10^−5^) were down-regulated for cells on HA surfaces compared to GB surfaces. Other pathways that were enriched with the down-regulated genes included alcoholism (KEGG: 05034; *P* = 6.89*P* × 10^−7^) and Systemic lupus erythematosus (KEGG: 05322; *P* = 5.64 × 10^−8^). Specifically, several genes from the H2 and H3 histone family that are common between the enriched pathways were down-regulated in osteoblasts on HA surfaces (Appendix A). A similar set of genes to those observed following ionic exposure, had higher expression on HA surfaces compared to both GB and Ti surfaces, including *BMP2*, *SPP1* and *ANKH* (Appendix A).

## 3. Discussion

The detrimental effect of prostheses-derived Co and Cr debris on the survival and function of bone cells is well established. However, little is known about the molecular mechanisms that mediate these effects. In this study, we used genome-wide microarrays to obtain a comprehensive insight into the response of primary human osteoblasts and osteoclasts at clinically relevant concentrations of 5 µg/L and 500 µg/L Co and Cr ions representing systemic and periprosthetic metal exposure in the setting of a well-functioning prosthesis, as well as CoCr nanoparticles. We also identify gene clusters and KEGG pathways for which the expression correlates with increasing Co^2+^:Cr^3+^ concentrations (Table 3). Furthermore, we describe the effects of routinely used grit-blasted, plasma-sprayed titanium and hydroxyapatite prosthetic surfaces on primary osteoblast gene expression without, and following, metal exposure.

### 3.1. Systemic Levels of Metal Exposure Inhibit Osteoclast Differentiation and Function

Total-body bone mineral density has been reported to be 5% higher in patients with well-functioning MoM hip resurfacing with median systemic concentrations of Co and Cr at 1.48 µg/L and 2.5 µg/L respectively, compared to those with conventional metal-on-polyethylene (MoP) prosthesis with >10 times lower metal ion levels [3]. This is accompanied by a decrease in serum markers of osteoclast and osteoblast activity suggesting reduced bone turnover. The results from this current study are consistent with these observations, and show down-regulation of genes associated with osteoclast differentiation and function in osteoclast precursors as well as mature osteoclasts at a comparable concentration of 5 µg/L Co^2+^:Cr^3+^. This, accompanied by lower transcription of genes associated with the focal adhesion and ECM interaction pathways that promote osteoclast differentiation via increasing RANK expression [13], further supports a reduction in osteoclast activity at systemic concentrations of metal ions.

The absence in alterations of gene expression in osteoblasts at this concentration suggests that the reduced systemic osteoblast activity observed in patients may be a result of de-coupling in osteoclast-mediated osteoblast function [14,15], and not a direct effect on osteoblasts themselves. This is further supported by previous reports that show no significant effect on osteoblast survival and function at equivalent concentrations [7]. Taken together, our data suggests that the systemic increased total-body bone mineral density may primarily be due to the direct inhibitory effects of metal ions on osteoclast differentiation and resorptive function.

### 3.2. Periprosthetic Levels of Metal Exposure Inhibit Differentiation and Function of Osteoblasts and Osteoclasts

At higher concentrations of 500 µg/L Co^2+^:Cr^3+^, equivalent to those observed in the periprosthetic milieu [16], increased expression of genes such as *PDK-1*, *VEGFA* and *BNIP3* that are associated with HIF-1 signaling pathway was observed in osteoblasts. Cobalt chloride is a well-recognized hypoxia mimetic that stabilizes HIF-1α and therefore this is an expected observation [17,18]. Nevertheless, the data is consistent with a recent clinical study that reported up-regulation of HIF-1α target genes in the periprosthetic tissue of patients with MoM hip replacement undergoing revision surgery [19]. HIF-1α stabilization seems to be a localized periprosthetic effect as a separate study did not observe any correlation in serum HIF1α levels in MoM patients with higher chronic metal exposure [20]. Additionally, the absence of an effect on HIF-1 signaling in osteoclast precursors or mature osteoclasts suggests that hypoxic signaling may be limited to cells of mesenchymal origin. This is supported by a previous study in which we observed lower uptake of Co^2+^ in mature osteoclasts compared to human osteoblast-like cells [21]. Furthermore, genes associated with fructose and mannose pathways (*PFKB3*, *PFKB4* and *ALDOC*, Table 2) that were up-regulated in osteoblasts are likely to be secondary to HIF-1α signaling, and represent a shift to a more glycolytic metabolic mode to sustain cellular energy needs, as shown previously [22,23]. The evidence for the effects of hypoxic signaling in osteoblasts is conflicting. Utting et al. (2006) show that hypoxic signaling inhibits differentiation and bone-forming capacity of primary rat osteoblasts [24]. Conversely, a study in murine models demonstrated that stabilization of HIF-1α and the shift to glycolytic metabolism increases osteoblast numbers and induces bone formation [25]. In our previous study, we observed that exposure to 500 µg/L Co^2+^:Cr^3+^ does increase osteoblast mineralization in vitro, without affecting cell survival [7]. However, the effect of hypoxic signaling on the quality of the mineralized matrix remains unclear. In a recent study, Stegen et al. (2019) show that while stabilization of HIF-1α increases collagen crosslinking and mineral deposition, the metabolism of collagen is altered leading to skeletal dysplasia [26].

A small proportion of patients with MoM hip replacement present with a local inflammatory reaction to metal debris referred to as ARMD, characterised by macrophage and lymphocytic infiltrate [27,28]. In osteoclast precursors, which are cells derived from the same myeloid lineage as macrophages, we observe an up-regulation of immune chemoattractants (*CCL1* and *CCL23*), and inflammatory cytokines (*IL-1β* and *IL-6*) from the chemokine-chemokine signaling pathway that may partially explain the mechanism for ARMD. While no studies have investigated the effect of combined CoCr metal exposure on gene expression in these cells, a previous study observed an up-regulation in cytokine and chemokine genes including *IL-1β* and *IL-6* in peripheral blood monocytes following Co^2+^ exposure alone [29]. Endocytosis and membrane trafficking were shown to be essential in fusion of osteoclast precursors as well as in maintaining resorption activity in mature osteoclasts [30,31]. Down-regulation of genes associated with endocytic and actin cytoskeletal pathways following ionic and particulate metal exposure suggests reduced osteoclast differentiation from its precursors in the periprosthetic environment. Additionally, a similar reduction in differentiation and function of mature osteoclasts is suggested with down-regulation of focal adhesion, endocytosis and PI3K-AKT pathways [13,32,33,34].

Taken together with the effect on osteoblasts, the data suggests that bone turnover may be inhibited in the periprosthetic microenvironment, with implications for osseointegration and eventual failure of the prosthesis. 

### 3.3. Pathways with a Dose-Dependent Effects to Metal Ions

In this study, we identified gene clusters for both osteoblasts and osteoclasts, with significant dose-dependent correlations with increasing concentrations of Co^2+^:Cr^3+^. The positive correlation for genes associated with HIF-1 signaling (such as *VEGFA*, *PDK1*, *ENO2* and *SLC2A1*) and Glycolysis/Gluconeogenesis pathways (*PGM1*, *TPI1* and *PGK1*) with metal exposure in osteoblasts confirm them as the primary response in these cells. This dose-dependent increase in the genes from the Glycolysis/Gluconeogenesis pathway is consistent with a recent study that showed an increase in glycolytic parameters including increased lactate production in macrophages exposed to Co^2+^ and Cr^3+^. The study also demonstrated that this shift to glycolysis is downstream of HIF-1α stabilization, an effect primarily attributed to Co^2+^ [18].

Similarly, genes from the cytokine-cytokine receptor interaction pathway (*CXCL4*, *CXCR5*, *IL13RA1*, *CD27*) in osteoclast precursors and focal adhesion pathways (*DIAPH1*, *PXN*, and *CCND1*) in mature osteoclasts positively correlated with metal exposure. The dose-dependent changes in the expression of these genes provides us with a possible gene-panel for which subtle changes in expression may serve as an early marker for CoCr metal toxicity.

### 3.4. Response of Osteoblasts to Prosthetic Surfaces and Metal Exposure

Prosthetic surfaces are routinely modified topographically or chemically to make them more osseoconductive and promote osseointegration. GB, Ti and HA coating are some of the most commonly used alterations of prosthetic surfaces. In a previous study, we reported increased osteoblast differentiation and mineralization on HA surfaces compared to GB and Ti surfaces, and a protective effect on their osteogenic activity in the presence of CoCr ions and nanoparticles [7]. Here we show that HA surfaces, in comparison to GB and Ti surfaces, consistently promote expression of *BMP2* in osteoblasts which is known to increase alkaline phosphatase activity and mineralization in an autocrine manner [35]. This increase in *BMP2* expression on HA surfaces seems to be a conserved response with studies showing its up-regulation in smooth muscle cells and periodontal ligament cells when cultured on HA surfaces [36,37].

Even with exposure to periprosthetic concentrations of CoCr ions and nanoparticles, *BMP2* and *FOSB* expression that mediate osteoblast differentiation, and *SPP1* (osteopontin) a marker for mature osteoblasts, were consistently up-regulated. This, accompanied by a reduction in genes associated with necroptosis (*HIST2H3A*, *HIST1H2AM*, and *HIST1H2AK*) and cell cycle progression (*CCNB2* and *CDCA2*), suggests that osteoblasts on HA surfaces possess a pro-survival and differentiation state compared to those grown on GB and Ti surfaces, with implications for osseointegration. Indeed, these results translate clinically. In the National Joint Registry for England and Wales, the largest joint replacement clinical audit in the world, the 5-year revision rates for hydroxyapatite-coated Birmingham and Adept hip resurfacing systems are 3.66% (95% CI; 3.42–3.92) and 4.52% (3.87–5.27) respectively, compared to 8.3% (6.93–9.93) for Conserve Plus and 5.47% (4.48–6.66) for the Durom system that use a plasma-sprayed titanium coating [38]. Finally, the up-regulation of glutamate receptor *GRIN3A* on GB surfaces compared to Ti surfaces may explain the increased mineralization observed on these surfaces in vitro [7]. Glutamate signaling was indeed shown to increase osteoblast differentiation and function [39], and the up-regulation of *GRIN3A* may partly be a result of differential cell adhesion to the different surface roughness of the two materials [40].

Our study also has limitations. While the high threshold for differentially expressed genes used in this study gives us confidence in our findings, validation of transcripts for select genes using RT-PCR, and the resultant proteins with western blotting, would be useful. Furthermore, investigating gene responses for osteoblasts and osteoclasts in isolation although vital to discern direct effects of metal exposure, discounts the complex inter-cellular coupling that exists between osteoblasts and osteoclasts to maintain bone homeostasis, and using co-culture systems or in silico modeling could help better understand tissue-level effects. Finally, the use of Co and Cr ion combinations at the ratio of 1:1 is an approximation of the clinical setting after hip resurfacing. This may differ for MoM total hip replacements and vary between patients.

In conclusion, this study provides a comprehensive mechanistic understanding of the direct effects prosthesis derived Co and Cr have on primary human osteoblasts and osteoclasts in both the systemic and local periprosthetic environment. Furthermore, for the first time, to the best of our knowledge, several gene clusters and pathways that are regulated in a dose-dependent manner in bone cells are identified here, which could serve as early markers for metal toxicity. Finally, we provide a mechanistic basis for the improved clinical performance of hydroxyapatite-coated prostheses.

## 4. Materials and Methods

### 4.1. Metal Ion Preparation and Treatment

Cobalt (II) hexahydrate (CoCl2.6H2O) and chromium (III) chloride hexahydrate (CrCl3.6H2O) (Fluka, Gillingham, UK) served as salts for Co^2+^ and Cr^3+^, respectively. For each metal salt, 0.2 M stock solutions were prepared in sterile water and serially diluted to 100× working concentrations. Subsequently, equal volumes of Co^2+^ and Cr^3+^ solutions (Co^2+^:Cr^3+^) were added to culture media to get the final treatment concentrations. The stability of these metal ions in culture media has been confirmed previously using flame–atomic absorption spectroscopy [4]. Control treatment contained equivalent volume of sterile distilled water to maintain conditions and are referred to as 0 µg/L treatments.

Co and Cr_2_O_3_ nanoparticles (a kind gift from Dr Ferdinand Lali, Imperial College, London, UK) were 40 nm and 30 nm in size respectively. Both osteoblasts and osteoclasts were treated with 100 nanoparticles each of Co and Cr_2_O_3_ per cell (CoCr nanoparticles), based on their seeding densities. Using the specific surface area (SSA) of the nanoparticles (50 m^2^/gm for Co, and 140 m^2^/gm for Cr_2_O_3_), and the surface area of a sphere (4πr^2^), we deduced the approximate particle number per gram of nanoparticles. These calculations were used to form a 1000× stock of the working concentration in 100% ethanol. Prior to treatments, the suspension was sonicated for 10 min to disaggregate the particles and diluted in osteogenic media to obtain the working concentration. Control treatment contained equivalent volume of ethanol to maintain conditions.

### 4.2. Primary Human Osteoblast Culture

Primary human osteoblasts were obtained from trabecular bone explants of patients undergoing joint replacement surgery following approval by South Yorkshire and Northern Derbyshire Musculoskeletal Biobank and approved by the National Research Ethics Service (15/SC/0132; Study reference: SYNDMB031), and all patients gave informed written consent prior to participation [41]. Briefly, the trabecular bone fragments were diced and washed thoroughly with PBS and cultured in petridishes with complete media containing DMEM GlutaMAX supplemented with 10% FBS, 100 U/mL penicillin, 100 µg/mL streptomycin and 2.5 µg/mL Amphotericin-B (Gibco, Paisley, UK). After a week, the media was replaced twice weekly for a further 4–6 weeks till the explant cultures reached confluence. The osteoblast-like phenotype of these cells was confirmed by assessing their alkaline phosphatase activity and their ability to mineralize in vitro, and was described previously [7]. Subsequently, cells were cultured in vitro on tissue culture plates, or on GB, Ti and HA coated prosthesis surfaces (supplied by JRI Orthopaedics Ltd., Sheffield, UK) that better mimic the periprosthetic physiological setting and were manufactured using the same processes as used commercially.

Osteoblasts grown in tissue culture plates were treated with a combination of 0, 5, and 500 µg/L of Co^2+^:Cr^3+^ to mimic the systemic concentrations observed following metal-on-metal hip resurfacing. For cells grown on prosthetic surface, periprosthetic concentrations of 1000 µg/L Co^2+^:Cr^3+^ were used. In both cases, the osteoblasts were exposed to the different concentrations of metal ions for 24 h. Previous clinical reports on the levels of circulating and periprothetic Co and Cr has informed the use of these concentrations in this study [42,43,44,45].

### 4.3. Primary Human Osteoclast Culture

Primary human osteoclasts were generated as described previously from human peripheral blood of healthy volunteers following research ethics committee approval [46]. Briefly, CD14+ enriched monocyte population was isolated from peripheral blood and seeded onto sterile dentine disks in osteoclastogenic media containing α-MEM GlutaMAX supplemented with 10% FBS, 100 U/mL penicillin, 100 µg/mL streptomycin (Gibco, Paisley, UK), 25 ng/mL M-CSF (R&D Systems, Abingdon, UK) and 30 ng/mL RANKL (R&D Systems, Abingdon, UK), and cultured in humidified incubator at 37 °C and 7% CO_2_. To investigate the effects of metal ions on developing osteoclasts, the metal ion treatments were added at day 3, and replaced every 2–3 days until the onset of resorption. To investigate the effects on mature osteoclasts, metal ion treatments were added for 24 h after the onset of resorption (typically day 14).

Both developing and mature osteoclasts were treated with 0, 5 and 500 µg/L of Co^2+^:Cr^3+^ to mimic systemic concentrations observed following metal-on-metal hip resurfacing. The developing osteoclasts were treated continuously til 24 h after the onset of resorption while mature osteoclasts were treated once for 24 h following the onset of resorption.

### 4.4. RNA Extraction and Gene Expression

Following exposure to metal ion treatments, the cells were washed twice with ice-cold PBS and total RNA extracted using the RNeasy Extractions Kits (Qiagen, Hilden, Germany) as per manufacturer’s protocol and the quality checked using the Agilent 2200 TapeStation (Agilent Technologies, Santa Clara, CA, USA). Gene expression profiling was performed using Agilent SureprintG3 Gene Expression Microarrays (8 × 60 K format) (Agilent Technologies, Santa Clara, CA, USA). Total RNA from was labeled with cyanine-3 CTP and hybridized to the array as per manufacturer’s instructions, the arrays read using the Agilent Microarray Scanner, and the data extracted with Feature Extraction v10.7 (Agilent Technologies, Santa Clara, CA, USA).

### 4.5. Data Analyses

The microarray data were imported into R using the ‘limma’ Bioconductor package for quality assessment and processing [47]. Three arrays were removed from the Osteoclasts dataset due to significant spatial artifacts present on the raw images. Background correction was performed using the normexp method in limma and signal between arrays was calibrated using quantile normalization. The normalized data were then filtered to the set of 50% most variable probes in order to increase the power to detect differential expression [48]. Differentially expressed probes were identified using a linear model approach followed by empirical Bayes shrinkage and the threshold was set at Log_2_FC > 1 and *P* < 0.001 [47]. DAVID (Ver. 6.8) was used to conduct KEGG enrichment analysis of differentially expressed genes [49,50]. The Benjamini-Hochberg corrected *p* value < 0.05 was set as the threshold for statistically significant enrichment. In a separate analysis, genes with an association between expression level and concentration were identified using the maSigPro Bioconductor package [51].

## Figures and Tables

**Figure 1 ijms-22-05225-f001:**
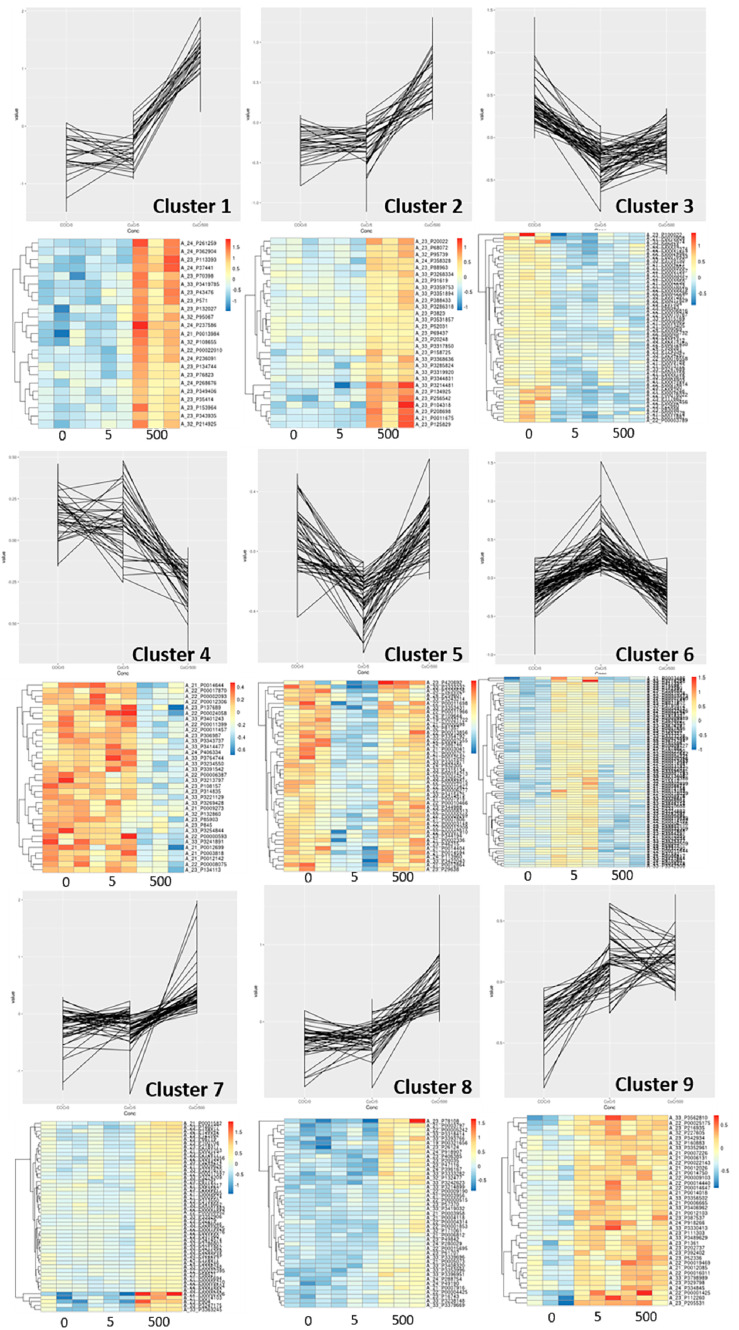
Expression profiles for osteoblast gene clusters with statistically significant expression changes with increasing concentrations of Co^2+^:Cr^3+^ (0, 5 and 500 µg/L). Corresponding heat map for each plot is illustrated underneath. Gene list for each cluster is available in Appendix A.

**Figure 2 ijms-22-05225-f002:**
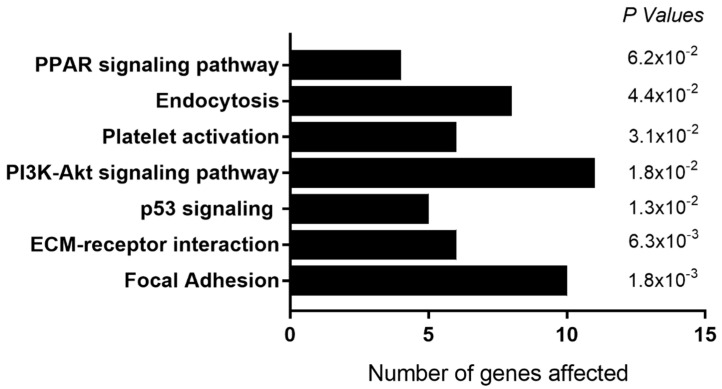
Functional classification of genes down-regulated in osteoclast precursors following exposure to 5 µg/L Co^2+^:Cr^3+^. KEGG pathways that were enriched in the differentially expressed genes, with the number of genes in each pathway and the corresponding *P* value are illustrated here.

**Figure 3 ijms-22-05225-f003:**
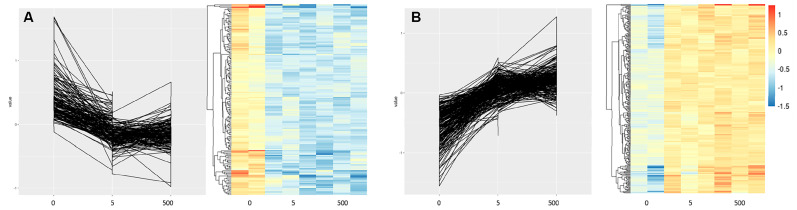
Expression profiles for osteoclast precursor gene clusters with statistically significant expression changes with increasing concentrations of Co^2+^:Cr^3+^ (0, 5 and 500 µg/L). Corresponding heat map for each plot is illustrated underneath. (**A**) Cluster 1 with genes that are overrepresented in calcium signaling and Ras signaling pathway, and associate negatively to metal concentrations, and (**B**) Cluster 4 have genes overrepresented in cytokine-cytokine receptor interaction pathway and have positive association with increasing concentrations of metal ions. Gene list for all eight clusters can be found in Appendix A.

**Table 1 ijms-22-05225-t001:** Genes altered in osteoblasts following 24 h exposure to 500 µg/L Co^2+^:Cr^3+^.

Gene	Log_2_FC	*p* Value
*PDK1*	2.04	1.16 × 10^−7^
*ANKRD37*	1.89	3.63 × 10^−5^
*APLN*	1.88	3.18 × 10^−5^
*AK4*	1.85	2.78 × 10^−4^
*BNIP3*	1.83	1.41 × 10^−6^
*PFKFB3*	1.69	6.57 × 10^−5^
*VEGFA*	1.59	3.85 × 10^−4^
*VLDLR*	1.59	8.95 × 10^−4^
*SLC2A1*	1.59	6.65 × 10^−4^
*PFKFB4*	1.59	2.32 × 10^−5^
*LOC101929947*	1.51	6.00 × 10^−4^
*TCAF2*	1.47	3.20 × 10^−4^
*ALDOC*	1.46	1.53 × 10^−4^
*ENO2*	1.42	2.98 × 10^−5^
*INHBB*	1.41	1.89 × 10^−4^
*FAM162A*	1.33	8.58 × 10^−6^
*P4HA1*	1.25	1.36 × 10^−4^
*DDIT4*	1.22	5.04 × 10^−4^
*NXPH4*	1.21	3.79 × 10^−4^
*EGLN1*	1.19	4.75 × 10^−5^
*GYS1*	1.17	4.17 × 10^−5^
*BNIP3L*	1.16	9.58 × 10^−6^
*PGK1*	1.12	7.77 × 10^−5^
*RORA*	1.10	8.68 × 10^−4^
*PPP1R3C*	1.01	1.44 × 10^−4^

**Table 2 ijms-22-05225-t002:** Functional classification of genes altered in osteoblasts following 24 h exposure to 500 µg/L Co^2+^:Cr^3+^.

KEGG Pathway	KEGG ID	Log_2_FC	Genes								
			*PDK1*	*PFKFB3*	*VEGFA*	*SLC2A1*	*PFKFB4*	*ALDOC*	*ENO2*	*EGLN1*	*PGK1*
**HIF Signalling**	**04066**	4.15 × 10^−7^									
Fructose and mannose metabolism	00051	8.9 × 10^−4^									
Renal cell carcinoma	05211	3.34 × 10^−2^									

**Table 3 ijms-22-05225-t003:** KEGG pathways that associate with increasing concentrations of Co^2+^:Cr^3+^.

Pathways	KEGG	Cell Type	Correlation	*p* Value
HIF1- signaling	04066	Osteoblasts	Positive	5.2 × 10^−7^
Glycolysis/Gluconeogenesis	00010	Osteoblasts	Positive	1.02 × 10^−6^
Ca^2+^ signaling	04020	Osteoclast precursors	Negative	1.3 × 10^−3^
Ras signaling	04014	Osteoclast precursors	Negative	3.6 × 10^−3^
Cytokine-cytokine receptor interaction	04060	Osteoclast precursors	Positive	4.7 × 10^−2^
Focal Adhesion	04510	Mature osteoclasts	Negative	5.0 × 10^−2^

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
