# Peer review of "Distinct Concentration-Dependent Molecular Pathways Regulate Bone Cell Responses to Cobalt and Chromium Exposure from Joint Replacement Prostheses"

_ijms, 2021, doi:10.3390/ijms22105225_

Round 1

Reviewer 1 Report

The manuscript entitled “Distinct Concentration-Dependent Molecular Pathways Regulate Bone Cell Responses to Cobalt and Chromium Exposure from Joint Replacement Prostheses” reports about analysis of whole genome microarrays to assess differential gene expression in primary human osteoblasts and osteoclasts in vitro, and in osteoblasts cultured on different prosthesis surfaces following exposure to Co and Cr. This work offers an original view on the topic using a genome microarrays technique. Moreover, charts and tables could be proven useful for researchers in the field.

I just have a few suggestions:

Methods.

- “Both osteoblasts and osteoclasts were treated with 100 nanoparticles each of Co and Cr2O3 per cell (CoCr nanoparticles), based on their seeding densities” is unclear. How the authors count 100 nanopartices? Please claryfie that.

- How the authors performed gene functional classification analysis? It should be added to the method section.

Results.

- Autors reported about not observing large-scale alterations in gene expression between osteoblasts on GB and Ti surfaces. "The only differential expression we observed was a downregulation in MX1 (Log2FC=-1.39; P=3.7e-3) and an upregulation in GRIN3A (Log2FC=-1.12; P=1.1e-3) expression in cells on GB surfaces compared to Ti surfaces.". What can be the possible explanation of this? What can be the impact of MX1 downregulation and GRIN3A upregulation? Authors should elaborate on that in the discussion section.

Discussion.

- “The detrimental effect of prostheses-derived Co and Cr debris on the survival and function of bone cells is well established.” Authors should at least mention how many cytotoxicity related genes were found in their analysis and compare it with the literature.

- “We identify several gene clusters and pathways that are regulated in a dose-dependent manner that could serve as early markers for metal toxicity.”Authors should specyfie more clearly those early markers in the text or table.

- Please declare the main novelty in the abstract and conclusion more clearly.

All considered, I would strongly recommend this article for publication in the journal “International Journal of Molecular Sciences”

Author Response

Please see the attachment for the comments to the reviewer.

Reviewer 2 Report

The paper by Wilkinson and coworkers describes the differential gene expression pathways found on cobalt and chromium exposed osteoblasts and osteoclasts in vitro.
The topic is intresting and clearly reported.
The manuscript is organized in a logical manner and is well written.
In this reviewer opinion, some points need revision:

- Please cite the relevant references supporting the choice of systemic and periprosthetic concentrations of Co and Cr ions, as well as of the nanoparticles.
- the abstract section should highlight the potential of the current study, i.e. the identification of a gene panel undergoing expression changes to be used as early marker of metal toxicity. 
- Lines 340-348: the limitations of this study are reported in an objective manner. It is suggested to add the lack of functional evaluations in vitro (i.e. matrix deposition was not investigated, as well as cell morphology and protein synthesis).
- Please provide details on the nanoparticles exploited during the study (e.g. particle size distribution)
- Osteoblast and osteoclast cells were isolated from patients. Osteoblast CD markers should have been reported to prove the identity of the isolated cells. At least, for osteoclast cells, CD14+ enrichment was recalled.

Author Response

(The authors gave the same response as above.)
